

# Comparing the simulated influence of biomass burning plumes on low-level clouds over the southeastern Atlantic under varying smoke conditions.

Alejandro Baró Pérez[1,2,a], Michael S. Diamond[3], Frida A.-M. Bender[1,2], Abhay Devasthale[4], Matthias Schwarz[5], Julien Savre[6], Juha Tonttila[7], Harri Kokkola[7], Hyunho Lee[8], David Painemal[9], and Annica M. L. Ekman[1,2]

[1]Department of Meteorology, Stockholm University, Stockholm, Sweden
[2]Bolin Centre for Climate Research, Stockholm, Sweden
[3]Department of Earth, Ocean, & Atmospheric Science, Florida State University, Tallahassee, FL, USA
[4]Meteorological Research Unit, Research and development, Swedish Meteorological and Hydrological Institute, Norrköping Sweden
[5]GeoSphere Austria, Hohe Warte 38,1190 Vienna, Austria
[6]Meteorological Institute, Fakultät für Physic, Ludwig-Maximilians-Universität Munich, Munich 80333, Germany
[7]Finnish Meteorological Institute, Kuopio, Finland
[8]Department of Atmospheric Science, Kongju National University, Gongju 32588, South Korea
[9]NASA Langley Research Center, Hampton, Virginia, USA
[a]now at: Department of Space, Earth and Environment, Chalmers University, Gothenburg 41296, Sweden

**Correspondence:** Alejandro Baró Pérez (baralej@chalmers.se)

**Abstract.** Biomass burning plumes are frequently transported over the Southeast Atlantic stratocumulus deck during the southern African fire season (June-October). The plumes bring large amounts of absorbing aerosols as well as enhanced moisture, which can trigger a rich set of aerosol-cloud-radiation interactions with climatic consequences that are still poorly understood. We use large-eddy simulation (LES) to explore and disentangle the individual impacts of aerosols and moisture on the under-

lying stratocumulus clouds, the marine boundary layer (MBL) evolution and the stratocumulus to cumulus transition (SCT) for three different meteorological situations over the Southeast Atlantic during August 2017. For all three cases, our LES shows that the SCT is driven by increased sea surface temperatures and cloud-top entrainment as the air is advected towards the equator. In the LES model, aerosol indirect effects, including impacts on drizzle production, have a small influence on the modeled cloud evolution and SCT, even when aerosol concentrations are lowered to background concentrations. In contrast,

local semi-direct effects, i.e aerosol absorption of solar radiation in the MBL, causes a reduction in cloud cover that can lead to a speed-up of the SCT, in particular during daytime and during broken cloud conditions, especially in highly polluted situations. The largest impact on the radiative budget comes from aerosol impacts on cloud albedo; the plume with absorbing aerosols produces a total average net radiative cooling effect between -4 and -9 $Wm^{-2}$ over the three days of simulations. We find that the moisture accompanying the aerosol plume produces an additional cooling effect that is about as large as the total

aerosol radiative effect. Overall, there is still a large uncertainty associated with the radiative and cloud evolution effects of biomass burning aerosols. A comparison between different models in a common framework, combined with constraints from in-situ observations, could help to reduce the uncertainty.



# 1 Introduction

Biomass burning aerosols (BBA) influence the Earth's weather and climate (Liu et al., 2020), but their net impact is difficult
to quantify because of the complex interactions between aerosols, radiation and clouds. The two main chemical components
of BBA are pure black carbon (BC) and organic carbon (OC); the former predominantly absorbs and the latter mostly scatters
solar radiation (Bond and Bergstrom, 2006). The combined absorption and scattering properties of BBA (direct aerosol effect)
modify the radiative fluxes in the atmosphere, leading to temperature changes, and in general, to an energy redistribution, that
affects atmospheric stability and alters cloud evolution and precipitation (semi-direct aerosol effect). Moreover, BBA can also
serve as cloud condensation nuclei (CCN) and ice nuclei (INP) and alter the radiative properties of clouds and their lifetime
(indirect effects, (Twomey, 1977; Albrecht, 1989)).

Several tropical regions on the planet experience biomass fires every year that release considerable amounts of BBA to the
atmosphere, affecting not only local but also remote areas (Herbert et al., 2021). The Southeast Atlantic (SEA) stands out as
one region receiving substantial quantities of BBA from June to October (De Graaf et al., 2020; Deaconu et al., 2019; Ichoku
et al., 2003). These aerosols originate from fires over the Southwestern African savanna that are transported westwards by the
predominant trade winds. In addition, the SEA hosts one of the largest semi-permanent stratocumulus (Sc) cloud decks on
the planet, providing an ideal environment for frequent interactions between BBA, Sc clouds, and radiation to occur. Another
notable characteristic of the SEA is the typical occurrence of stratocumulus to cumulus transitions (SCTs) when air masses flow
towards the equator on the eastern side of the semi-permanent South Atlantic subtropical anticyclone (Wood, 2012). The so-
called deepening-warming mechanism (Bretherton and Wyant, 1997) suggests that SCTs are primarily driven by increasing sea
surface temperatures (SSTs) as the air moves from the subtropics to the tropics. Warmer SSTs lead to increased surface latent
heat fluxes that are positively buoyant and intensify the turbulent kinetic energy (TKE) in the boundary layer. The TKE increase
strengthens the cloud-top entrainment which deepens the boundary layer. However, the entrained air is positively buoyant and
therefore reduces the mixing between the cloud and the sub-cloud layers. The latter eventually leads to a decoupled boundary
layer, where the flux of surface moisture to the Sc cloud is diminished, and the formation of cumulus clouds below the Sc is
favored. According to this theory, the speed of the SCT is mostly modulated by the strength of the MBL-capping inversion
whereas precipitation formation is less important (Sandu and Stevens, 2011). However, several recent studies have found that
drizzle formation may indeed be highly relevant for SCTs, especially when aerosol concentrations are low, and when feedbacks
between aerosols, cloud droplet number, and precipitation are considered (Yamaguchi et al., 2015, 2017; Erfani et al., 2022;
Diamond et al., 2022).

The BBA are frequently transported in the free troposphere (FT) over the SEA, above the Sc-topped MBL. The combined
direct and semi-direct effects in the FT can impact the Sc and MBL, mostly when the distance between the aerosol plume and
the clouds is small (Herbert et al., 2020; Baró Pérez et al., 2021; Diamond et al., 2022). For instance, the absorption of SW
radiation by the aerosols can warm the FT and strengthen the inversion at the top of the MBL, slowing down its deepening.
Diamond et al. (2022) found that the SW absorption in the FT also can reduce the subsidence rate, which can modulate the
timing for a potential contact between the aerosol plume and the underlying Sc clouds. Studies based on satellite data have



also shown an increase in cloud cover and cloud thickness in the presence of BBA in the FT (Wilcox, 2010; Costantino and Bréon, 2013). The BBA plumes over the SEA are typically accompanied by enhanced moisture originating from the continental boundary layer (Haywood et al., 2004; Adebiyi et al., 2015; Zhou et al., 2017; Deaconu et al., 2019). The enhanced moisture

transport is not directly caused by the biomass burning itself but happens to coincide with the transport of BBA (Pistone et al., 2021). Water vapor absorbs mainly at near-infrared and infrared wavelengths longer than 0.7 $\mu$m (Ramaswamy and Freidenreich, 1991; Collins et al., 2006). Thus, by increasing the downward longwave (LW) fluxes, moisture associated with BBA in the free troposphere can suppress MBL deepening (Eastman and Wood, 2018) by reducing the net Sc top LW cooling, as has been shown by large-eddy simulations (e.g.Yamaguchi et al. (2015), Zhou et al. (2017)).

The BBA can also entrain and mix into the MBL, affecting clouds, precipitation, and the MBL evolution through aerosol indirect and semi-direct effects. The indirect effects resulting from a BBA plume can be manifested as a chain of processes. Following Twomey (1977) and Albrecht (1989), an increase of the cloud droplet number concentration ($N_c$) can result in a reduction of cloud droplet size, leading to a higher cloud albedo and eventually precipitation suppression. This can potentially extend the cloud's lifetime and increase the cloud depth and liquid water path (LWP). However, precipitation suppression can

also increase cloud-top radiative cooling, leading to an increase in the entrainment rate and a reduction of the LWP (Wood, 2012; Gryspeerdt et al., 2019). The semi-direct effect of BBA within the MBL can manifest itself as a temperature increase that leads to a reduction of the relative humidity (RH) and, consequently, a decrease in cloudiness (Hansen et al., 1997; Ackerman et al., 2000; Diamond et al., 2022). In addition to the above, the moisture associated with BBA plumes can result in that relatively humid air is entrained into the MBL, which can lead to an increase in cloud cover with increasing entrainment,

instead of the opposite that typically would occur for a clean, dry free troposphere (Eastman and Wood, 2018).

The aerosol amount and humidity tend to co-vary within BBA-moist plumes, and their individual effects on MBL evolution and Sc clouds are therefore difficult to disentangle using observational data, e.g. from satellites (Baró Pérez et al., 2021). Their relative impacts might also vary depending on the meteorological situation and the magnitude of the perturbations (Pistone et al., 2016). Therefore, a modeling perspective may be useful for examining the issue. Yamaguchi et al. (2015)

used Lagrangian large-eddy simulations and an idealized SCT case (Sandu and Stevens, 2011) to explore how SCTs can be influenced by a plume of enhanced moisture and smoke. They found that cloud-top entrainment and MBL deepening were reduced when the smoke layer was located above the Sc deck, due to smoke absorption and a strengthening of the inversion. When the plume entrained into the MBL, drizzle was suppressed, which together with the enhanced moisture associated with the aerosol plume contributed to Sc cloud sustenance and a delay in the SCT. With some modifications, Zhou et al. (2017) also

used the Sandu and Stevens (2011) SCT case study. However, they obtained an acceleration of the SCT when the aerosol plume made contact with the Sc deck, due to an increased $N_c$, smaller droplets, more evaporative cooling, and enhanced cloud-top entrainment. Note that the simulations by Zhou et al. (2017) did not include prognostic aerosol concentrations, i.e. their model setup would not be able to produce any drizzle-driven acceleration of the SCT, which could explain some of the differences compared to Yamaguchi et al. (2015).

One disadvantage of the studies by Yamaguchi et al. (2015) and Zhou et al. (2017) is that they both used idealized meteorological conditions, representative of the Northeastern Pacific. In contrast, Diamond et al. (2022) used data from the joint



ObseRvations of Aerosols above CLouds and their intEracionS (ORACLES)-CLouds and Aerosol Radiative Impacts and Forcing: Year 2017 (CLARIFY) campaigns to simulate an SCT case over the SEA. In their simulations, they also incorporated the effect of smoke on the large-scale circulation by forcing their LES model with output from a regional climate model. Diamond et al. (2022) found that the large-scale thermodynamic and dynamic adjustments, which cannot be explicitly simulated by an LES model, had the largest impact on the SCT except when aerosol concentrations were very low. This delay was to some extent counteracted by local (within the MBL) semi-direct effects that decreased cloud cover. However, similar to Yamaguchi et al. (2015) and Zhou et al. (2017), Diamond et al. (2022) found that local semi-direct effects did not dominate the impact of BBA on cloud evolution.

To summarize, BBA plumes and associated moisture (either overlying or mixed into the MBL) can influence the MBL, Sc clouds, and in consequence SCTs over the SEA in multiple and sometimes counteracting ways. The complexity of the interactions has caused disagreement between previous modeling studies using large-eddy simulation. Most of these studies were based on idealized meteorology. Only one study has examined conditions representative of the SEA and they focused on only one specific case (Diamond et al., 2022). Since the location, timing, and levels of pollution of the BBA plumes influence the clouds and MBL evolution, the analysis of different situations can give a wider perspective of the possible ways in which the humid BBA plumes can affect low-level clouds over the SEA.

In this work, we use the MISU-MIT Cloud and Aerosol (MIMICA) LES model (Savre et al., 2014) to simulate stratocumulus-topped boundary layers and SCTs for three different meteorological situations over the SEA, during August 2017, characterized by the presence of moist absorbing aerosol plumes interacting with the MBL. The three situations are chosen to study cases that are clearly different in the levels of pollution and moisture in the FT and also with respect to when the BBA plume appears in the FT, and when it mixes into the MBL. We explore the individual influence of aerosols and moisture on the diel (24 hours) cycle of the MBL, the Sc clouds, and on the SCT. Furthermore, we compare the impacts observed in the three situations and investigate the overall radiative effects of the absorbing aerosols as well as the enhanced moisture within the BBA plume. In Section 2, we describe the model setup and define some variables and parameters used in the analysis. Section 3 describes the results followed by a discussion and conclusions in Section 4

## 2 Methods

### 2.1 Model setup and simulations

The MIMICA LES code (Savre et al., 2014) solves a set of anelastic, non-hydrostatic governing equations. The model uses a two-moment bulk microphysics scheme (Seifert and Beheng, 2001, 2006) where the mass mixing ratios and the number densities of five hydrometeor types (i.e. cloud droplets, raindrops, ice crystals, snow and graupel) are defined as prognostic variables. In this work, we only use the microphysics associated with warm (no-ice) clouds and precipitation and use a radius of 25 $\mu$m to separate cloud droplets from raindrops. Hydrometeor size distributions are defined by gamma functions. The model is coupled to a version of the four-stream Fu-Liou-Gu radiative transfer model (Fu and Liou, 1993; Fu et al., 1997; Gu et al., 2003) which includes 6 bands for shortwave and 12 for longwave radiation. A two-moment aerosol module is used to



characterise the aerosol population (Ekman et al., 2006), with the possibility to define several log-normally distributed aerosol modes with different composition and size. Each aerosol mode is composed of a single or a combination of four aerosol types (black carbon, organic carbon, sulfate, and sea salt). Aerosols can act as CCN following the ($\kappa$)–Köhler theory (Petters and Kreidenweis, 2007) and are lost through cloud processing and precipitation. However, the model tracks the mass of activated aerosols and generates one aerosol from each evaporated droplet.

To perform the simulations, we have added explicit aerosol-radiation interactions to MIMICA. The implementation is an adaptation of the aerosol-radiation interactions used by Slater et al. (2020) in UCLA-LES-SALSA (Tonttila et al., 2017). MIMICA and UCLA-LES-SALSA share the same radiative transfer model (Fu and Liou, 1993; Fu et al., 1997; Gu et al., 2003), but in MIMICA the optical properties (optical thickness, single scattering albedo and phase function) are estimated for aerosol modes whereas in UCLA-LES-SALSA they are estimated for aerosol bins. The real and imaginary refractive indices

for each aerosol component are obtained from Hess et al. (1998). The mean refractive index of the aerosol mode will be proportional to the volume fraction of each aerosol type.

For our simulations, we use a 9.6 by 9.6 km horizontal domain with a resolution of 50 m. The vertical grid consists of 288 grid points from the surface to 6.5 km with a resolution of 10 meters below 2.5 km and a vertical stretching above (vertical distance between grid points increases by 10% each level). Aerosol properties described above are based on ORACLES-2017

measurements and follow Diamond et al. (2022). The BBA is represented by using a combination of black carbon (6.8%) and organic carbon (93.2%) in a single mode; the single scattering albedo (SSA) of this internal mixture is approximately equal to 0.85. The initial geometrical mean diameter of the aerosol distribution is 185 nm with a fixed geometric standard deviation of 1.5. The hygroscopicity parameter ($\kappa$) is set to 0.2, consistent with Fanourgakis et al. (2019); Howell et al. (2021); Diamond et al. (2022). We use a constant surface aerosol source of $70 cm^{-2} s^{-1}$ to maintain the background aerosol number

concentration ($N_a$) within reasonable values (Wang et al., 2010; Yamaguchi et al., 2015). For simplicity, the aerosol source is assumed to consist only of BBA, as in Yamaguchi et al. (2015) and Diamond et al. (2022). The horizontal wind divergence rate is set to $2.16 \cdot 10^{-6} s^{-1}$.

The simulations are initialized (in the entire model domain) and later forced (only in the free troposphere) with meteorological fields and aerosol conditions given by trajectories calculated with the Goddard Earth Observing System Model Version 5

(GEOS-5) (Molod et al., 2012), that are a subset of those computed in Painemal et al. (2018). In the trajectories, the parcels' initial longitudes span from 0 to 12°E, while their initial latitude is set at 25°S. A fixed vertical level of 250 m for the parcels is assumed, disregarding any vertical movement. At a specific point, the horizontal wind velocities are calculated by linearly interpolating values from 16 neighboring spatiotemporal (x, y, z, and time) points. To calculate the incremental changes in the parcels' locations after each integration on the latitude/longitude coordinate, equatorial and polar radii of 6378.137 km and

6356.752 km, respectively are used. The model time step is 10 minutes in GEOS-5, and the simple forward Euler method is used for time integration. When the parcels approach land too closely to obtain the 250-m winds the integration stops.

During MIMICA's simulations, the free troposphere is continuously nudged to the forcing (from GEOS-5 trajectories) temperature, $N_a$, and mass fraction of water in air (water vapor mixing ratio) on a timescale of 30 minutes, and on a timescale of 3 hours to the horizontal winds. Note that this setup means that the meteorological fields from the GEOS-5 forcing data will



**Table 1.** Trajectories used for the model simulations.

| Trajectory ID | Start date | Start time | Latitude |
|---|---|---|---|
| AUG03 | 2017-08-03 | 21:00 | 11.11 S |
| AUG16 | 2017-08-16 | 21:00 | 11.60 S |
| AUG31 | 2017-08-31 | 21:00 | 11.10 S |

always include FT semi-direct aerosol effects, as there are no GEOS-5 simulations where aerosol radiative effects are turned off (Diamond et al., 2022). We define the nudging base as the maximum inversion height in the model domain plus 100 m, and calculate the inversion height (or FT base) as the maximum vertical gradient of the potential temperature in each model column. The model is also forced with sea surface temperature (SST) values from GEOS-5 (Figure 1a).

We use three individual trajectories (cases) to force the LES, corresponding to periods starting on three different days (3rd,
16th and 31st) of August 2017 (see Table 1 and Figure 1b). The motivation to select these days are explained in sections 1 and 3.1. For each case, four simulations are carried out: the control (CTRL) with the original values for each variable in the trajectory, an experiment called N100 with a fixed $N_a = 100 mg^{-1}$ in the FT (this $N_a$ value is among the lowest in the FT for AUG03, which is the cleanest among the three cases, see Section 3.1), an experiment with no aerosol-radiation-interactions (Aer-rad-off), an experiment where the water vapor mixing ratio ($Q_v$) in the FT was reduced to values between 0.1 and
$0.4 g \cdot kg^{-1}$ (DRY simulation). These range of $Q_v$ values occur in AUG03 CTRL at the beginning of the simulation when the FT is relatively dry. In total, we have 12 different simulations. In the experiments corresponding to the same day, the FT is always nudged (above the nudging base) towards the same forcing values of temperature and horizontal winds. Thus, changes in $N_a$ (N100 experiment) or moisture (DRY experiment) or turning off the aerosol scattering and absorption (Aer-rad-off) are not going to affect the temperature and winds in the FT above the nudging base. This type of model setup also means that
aerosol-radiation interactions, and associated cloud adjustments, will only be fully effective below the nudging base (the MBL and the lower 100 m of the FT).

The differences between the Aer-rad-off and the CTRL experiments are used to evaluate the direct and associated semi-direct aerosol effects in the MBL. Contrasts between the DRY and CTRL experiments are a consequence of the radiative impact of moisture in the FT on the MBL and the moisture entrainment into the MBL. The comparison between the N100 and CTRL
experiments shows differences due to a combination of the indirect aerosol effect (due to changes in $N_a$) and the semi-direct aerosol effect (because lower $N_a$ produces less heating). By comparing Aer-rad-off and N100, we can also obtain an estimate of the indirect aerosol effect.

## 2.2 Parameters and variables used in the analysis

Here we briefly define some parameters and variables that we use in the analysis:

– **Cloud cover** is defined as the fraction of model columns with $LWP > 0.01 kg \cdot m^{-2}$ (Sandu and Stevens, 2011; Zhou et al., 2017).



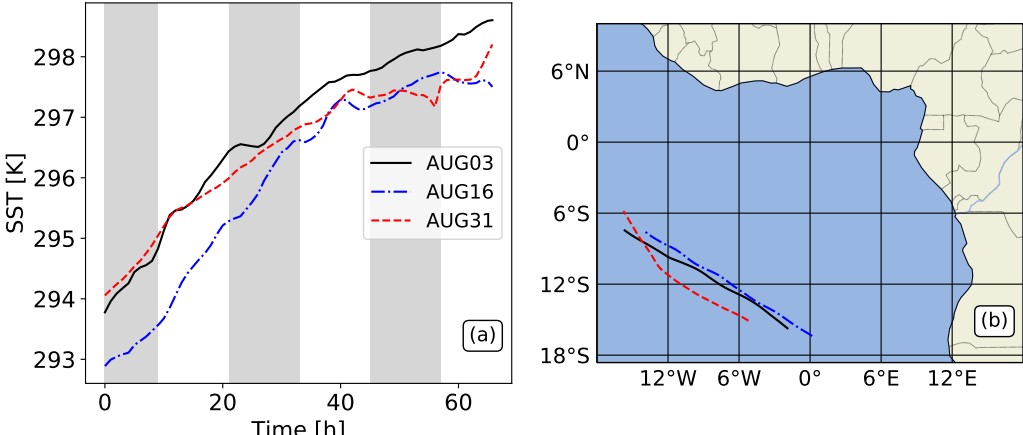

**Figure 1.** (a) Time series of sea surface temperature (SST) along the Lagrangian trajectories from GEOS-5, (b) Geographical location of the Lagrangian trajectories from GEOS-5

– **Vertically resolved cloud cover** is the fraction of total columns at each model level with cloud droplet mixing ratio $Q_c > 0.001 g \cdot kg^{-1}$ (following Zhou et al. (2017), who call this variable cloud fraction).

– **Marine boundary layer turbulent kinetic energy (MBL TKE)** is the domain-averaged TKE between the surface and the height of the inversion capping the MBL.

– **The stratocumulus to cumulus transition (SCT)** is defined as the time at which the cloud cover first decreases to half of its initial value (Sandu and Stevens, 2011).

– **The decoupling parameter** ($\delta Q_t$) is calculated as the difference between the total water mixing ratio ($Q_t$) from the bottom and the top 25% of the MBL, according to Jones et al. (2011) and Diamond et al. (2022).

– **Aerosol radiative effect at the top of the model domain** is calculated as:

   – $F_{CTRL} - F_{Aer-rad-off}$ due to direct and semi-direct aerosol effects

   – $F_{Aer-rad-off} - F_{N100}$ due to aerosol indirect effects (negligible semi-direct aerosol effect)

   – $F_{CTRL} - F_{N100}$ due to all aerosol effects combined.

   where $F$ is the net incoming SW or LW radiation.

– **Aerosol clear sky radiative effect at the top of the model domain (TOA)** is derived from $FCS_{Aer-rad-off} - FCS_{CTRL}$, where $FCS$ is the net outgoing SW or LW radiation with clear sky (no clouds).



- **The radiative effect due to the enhanced moisture in the aerosol plume** is calculated as the difference between the outgoing radiative fluxes in the DRY and the CTRL experiments: $F_{CTRL} - F_{DRY}$.

- **The cloud radiative effect (CRE)** at the top of the model domain is calculated as the difference between the upwelling clear sky SW (or LW) fluxes and the upwelling all sky SW (or LW) fluxes.

- **The entrainment rate** is calculated, following Lock (2001) and Bulatovic et al. (2021), as the difference between the subsidence (large-scale divergence rate) at the top of the inversion and the change in the inversion height with time.

- **Entrainment fluxes of aerosol and moisture** at the top of the MBL are calculated by multiplying $N_a$ and the mixing ratio of specific moisture ($Q_v$) with the entrainment rate, respectively.

## 2.3 Observations

We evaluated the simulated time evolution and diel cycle of cloud cover and liquid water path using geostationalry satellite sensor data. More specifically, we used cloud cover and LWP from SEVIRI retrieved by the NASA Langley Research Center using the Satellite Cloud and Radiation Property retrieval System algorithms (Painemal et al., 2015, 2012; Minnis et al., 2008). Both variables are averaged in boxes of 0.185° diameter around each data point in the GEOS-5 trajectories. LWP retrievals are screened to only include data with solar zenith angles below 70° and cloud fractions above 90%. We complement this information with retrievals from the latest (third) edition of CM SAF CLoud property dAtAset using SEVIRI (CLAAS-3, Meirink et al. (2022)). CLAAS-3 provides high-resolution (0.05 degrees in space and 15 minutes in time) cloud property retrievals, which is useful when evaluating the model performance in terms of simulating the time evolution of clouds.

## 3 Results

### 3.1 Control simulations

As highlighted in section 1, the three cases analysed (AUG03, AUG16, AUG31) differ in the levels of pollution and moisture in the FT, and in terms of when the BBA plume appears in the FT and when it mixes into the MBL (see Figures 2, 3, 4). Nevertheless, there are some characteristics that are common for all simulated CTRL cases. First, there is a visible covariance between $N_a$ (Figure 2) and specific humidity (Figure 3) within the FT BBA plumes, as typically happens over the SEA (see section 1). Second, some well-known features of the MBL diel cycle can be observed in all three cases: during nighttime, there is an increase in the cloud cover (Figure 5), vertically resolved cloud cover (Figure 4, LW cooling at the top of the MBL (Figure 6 a-c), and MBL TKE (Figure 6 d-f). There is also a fast increase in the height of the MBL inversion with time (Figure 6 j-l). During daytime, the cloud cover and liquid water path (Figure 5) decrease at the same time as there is a reduction of the MBL TKE and a slower deepening of the MBL compared to nighttime conditions. Cumulus start to develop under the Sc after some time in the simulations: around sunrise of the second day in AUG03, near the beginning of the simulation in AUG16, and after sunset of the first day in AUG31 (Figure 4). There is a general increase of the MBL decoupling during or after the





second night (Figure 6 m-o). The second day is characterized by a sharp decrease of the cloud cover and the vertically resolved cloud cover in all CTRL cases. The SCT, according to the criteria defined in Section 2.2, happens on the second day in all CTRL simulations; around midday in AUG03 and AUG16, and in the afternoon in AUG31. Precipitation is small in all CTRL

cases (Figure 7g-i), which suggests that, for these cases, drizzle formation does not have a substantial impact on the SCT in MIMICA. The aforementioned description of the MBL evolution in the three situations is consistent with a deepening-warming type of SCT, primarily driven by the SST increase (Figure 1a). The diurnal CRE is dominated by the SW fluxes (Figures 8a-c and Table 2). The domain-average CRE (Table 2) becomes less negative with each day of simulation as a result of the reduction of cloudiness.

Despite the similarities, there are also clear differences between the three cases. AUG03 is initially characterized by a relatively clean FT (Figure 2). Around midday of the second day and until the end of the simulation, the BBA plume becomes apparent in the FT, approximately between an altitude of 4000 m and the top of the MBL. The maximum values of the domain average $N_a$ in the plume are around $600 mg^{-1}$, and are reached towards the end of the simulation near the MBL top. The entrainment of aerosol in the MBL remains relatively low during all the simulation compared to AUG16 and AUG31 (Figure

9a-c).

AUG16 starts with a strong and humid aerosol plume in the FT with average $N_a > 3000 mg^{-1}$ near the MBL top. The BBA plume is in contact with the MBL top already at the beginning of the simulation and eventually entrains into the MBL, resulting in a relatively clean FT and a very polluted MBL (with $N_a > 1000 mg^{-1}$) during the second half of the simulation. This situation differs from the cases analyzed by Yamaguchi et al. (2015), Zhou et al. (2017), and Diamond et al. (2022) in

which the absorbing aerosol layer in the FT is initially clearly separated from the cloud layer, and only after a certain time makes contact with the Sc and entrains into the MBL.

In AUG31, the FT just above the MBL top remains relatively polluted and humid all the time, with $N_a$ values above $1000 mg^{-1}$. There is a gradual increase of $N_a$ with time in the MBL as a consequence of the entrainment of the aerosol plume (Figure 9c). This entrainment is, however, not as strong as the one in AUG16 (Figure 9b), and at the end of the simulation most

of the pollution remains above the MBL.

### 3.2 Comparison between the control simulations and the observations

Figure 5 compares cloud cover and liquid water path along the trajectories between SEVIRI (NASA) and the MIMICA simulations. In the case of SEVIRI (NASA), for each of the days (AUG03, AUG16 and AUG31), we averaged three contiguous trajectories that are expected to have identical thermodynamic profiles (e.g. the trajectory matching the simulation AUG03

and two other trajectories from SEVIRI (NASA) that are relatively close). We did this in order to reduce the lack of satellite observations due to the filtering applied to the dataset.

The SEVIRI (NASA) retrievals of cloud cover suggest a predominance of overcast conditions at the beginning of the three trajectories that later alternate with more broken clouds conditions (Figure 5 a-c). MIMICA also simulates cloud conditions along the trajectories that initially are overcast and later evolve into more broken cloud scenes. The sparse LWP data retrieved

from SEVIRI (NASA) has the best agreement with the LWP simulated by MIMICA in AUG03, while the biggest differences



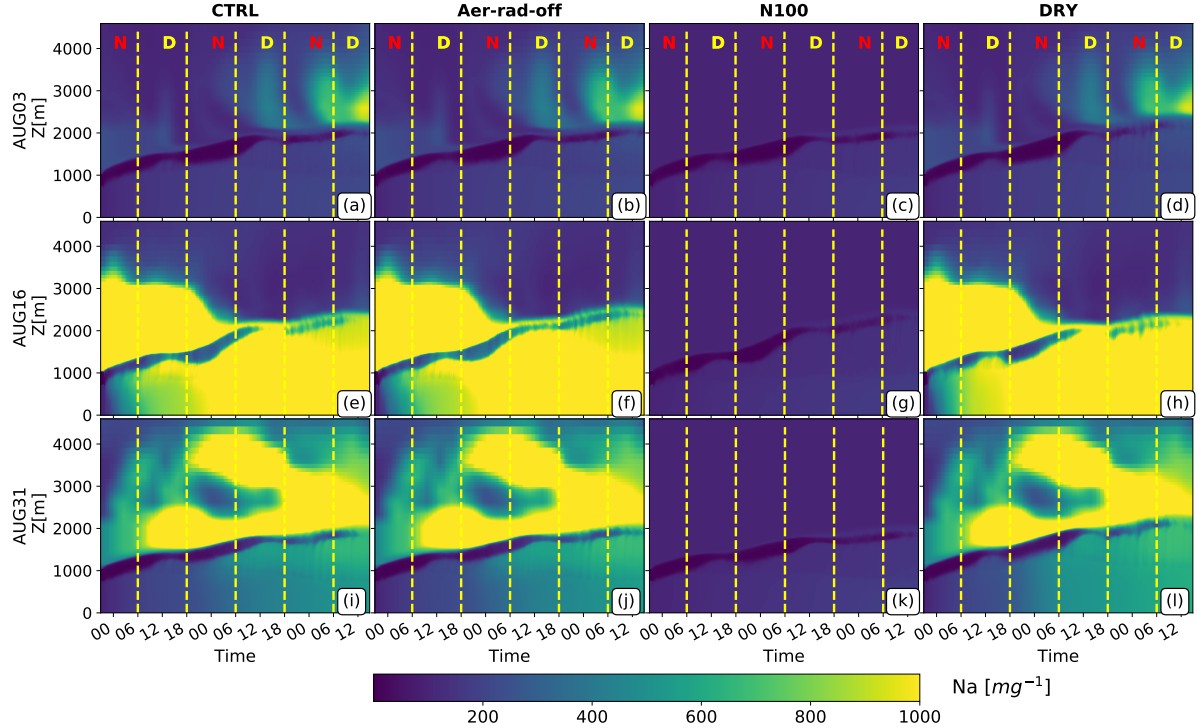

**Figure 2.** Temporal evolution of simulated profiles of the horizontally-averaged aerosol number concentration ($N_a$) from MIMICA. Results are shown for all experiments (CTRL, Aer-rad-off, N100 and DRY) and all cases: AUG03 (a-d), AUG16 (e-h), and AUG31 (i-l). Dashed lines separate nighttime (N in the upper subplots) from daytime (D). Z is the altitude.

are observed during the first day of AUG31. The comparison may be affected by the fact that our retrieval samples (SEVIRI) have a few values per time.

In order to obtain a more robust statistical evaluation, we compare diel cycles averaged during August over 10 years of data of cloud cover and LWP from CLAAS-3 with the equivalent variables for each of the trajectories in the MIMICA simulations
(Figure 10). There is a clear diel cycle of cloudiness and LWP in both the CLAAS-3 dataset and the MIMICA simulations. As expected, the cloudiness and the LWP are at minimum in the early afternoon. The cloud cover peaks early in the morning and in late evening in both the observations and the simulations. However, the cloud cover reaches its minimum a few hours earlier in the simulations compared to the observations and the amplitude of the diel cycle is stronger. The values of LWP are higher in the CLAAS-3 dataset than in MIMICA. Given the inherent differences in resolution between the CLAAS-3 retrievals and
the MIMICA simulations, we find the model produces reasonable results.





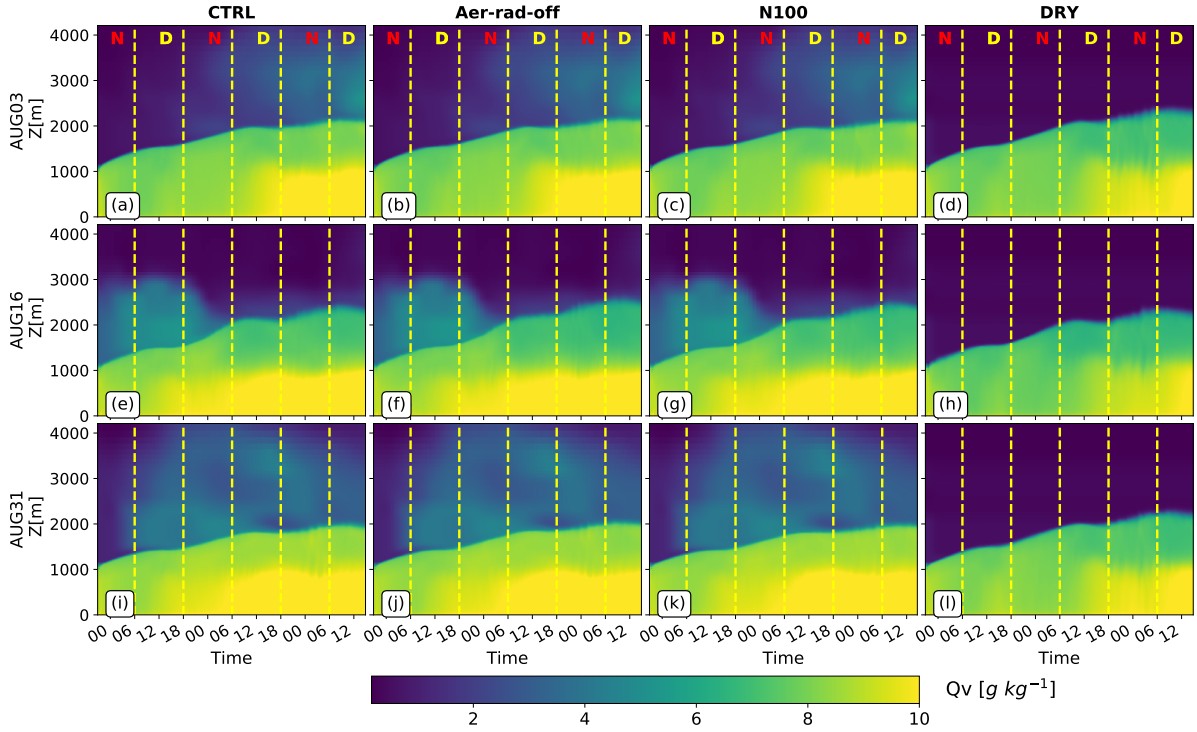

**Figure 3.** As in Figure 3 but for the water vapor mixing ratio ($Q_v$). Simulations are shown for: AUG03 (a-d), AUG16 (e-h), and AUG31 (i-l).

## 3.3 Influence of direct and semi-direct aerosol effects

In this subsection, we compare the Aer-rad-off simulations with the corresponding CTRL cases to study the influence of aerosol direct and semi-direct effects on the simulated cloud properties. Above-cloud semi-direct effects are excluded from the comparison because the forcing conditions in the FT are the same in both cases. For AUG03, the pollution levels are relatively

low and mainly present during the second half of the simulation. In addition, most of the BBA plume remains above the Sc deck and the MBL (Figure 2). Thus, any impacts associated with the semi-direct effect are relatively weak and mainly noticeable during or after the second day. The main feature when comparing CTRL (including aerosol-radiation interactions) with Aer-rad-off (without aerosol-radiation interactions) is a reduction in cloud cover during the second day when aerosol-radiation interactions are turned on (Figure 5 (d-f)). The daytime average difference in SW heating between CTRL and Aer-rad-off is

generally small during the second day (Figure 11b). However, a closer inspection (not shown) suggests that the aerosol SW heating in CTRL causes a temperature increase and a relative humidity decrease that most likely explains the reduction of cloudiness (initially the temperature increase only occurs above the MBL top, but later increases within the MBL as the clouds



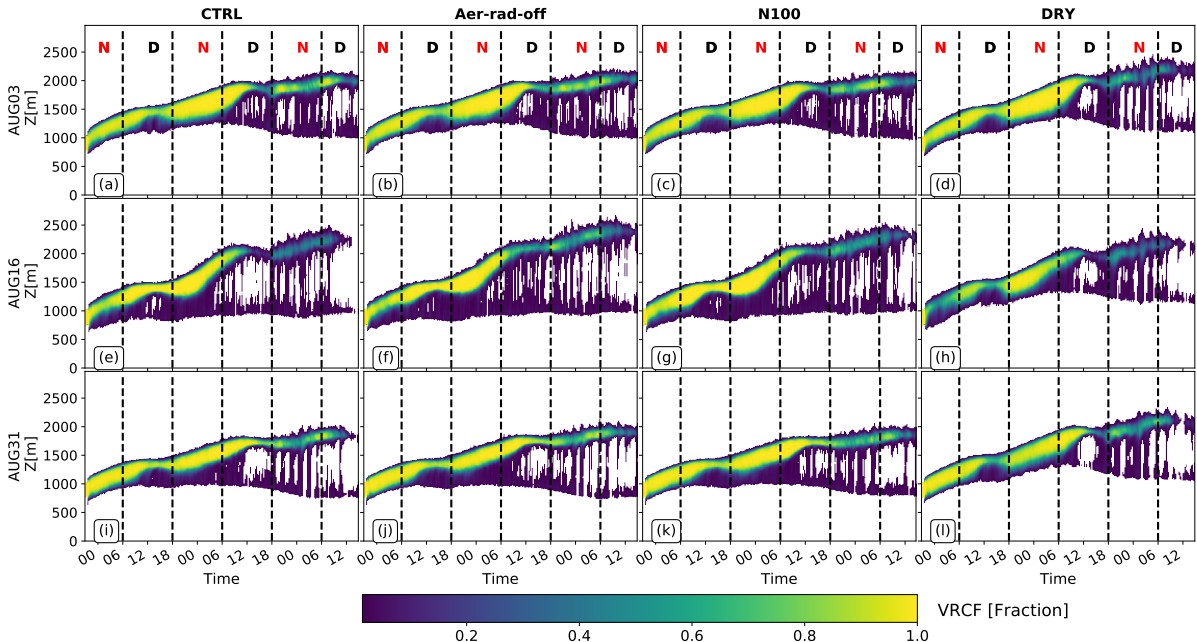

**Figure 4.** As in Figure 2 but for the vertically resolved cloud cover (VRCC). Simulations are shown for: AUG03 (a-d), AUG16 (e-h), and AUG31 (i-l).

break up). The diurnal average CRE (driven mostly by the SW fluxes) is reduced in absolute values in CTRL compared to Aer-rad-off during all three days, but more clearly during the last two days (Table 2). This reduction can be explained by a

combination of two factors: the decrease in cloud cover in CTRL compared to Aer-rad-off, which reduces the albedo, and the reduction of contrast in albedo between clear and cloudy skies (which makes the clear sky scenes brighter in CTRL compared to Aer-rad-off because the aerosols have a higher albedo than the sea surface). Differences in precipitation are small between CTRL and Aer-rad-off (Figure 7).

AUG16 shows the clearest impacts among all cases of aerosol-radiation interactions on simulated cloud properties and MBL

evolution due to the high levels of pollution (Figure 2). During the first day, the BBA plume is in contact with the Sc deck and there is also a substantial amount of pollution present within the MBL. However, the average daytime SW heating within the MBL does not differ substantially between CTRL and Aer-rad-off (Figure 11d) due to the overcast conditions. Consequently, the differences in cloud parameters between CTRL and Aer-rad-off (e.g. cloud cover and LWP, Figure 5) are small during the first day of simulation. The differences between CTRL and Aer-rad-off become more pronounced after the second night, when

the aerosol plume is mainly located within the MBL (Figure 2). The general reduction in cloud cover during the second day (see Section 2.1) allows more solar radiation to penetrate the MBL, causing a strengthening of the direct and semi-direct effects of



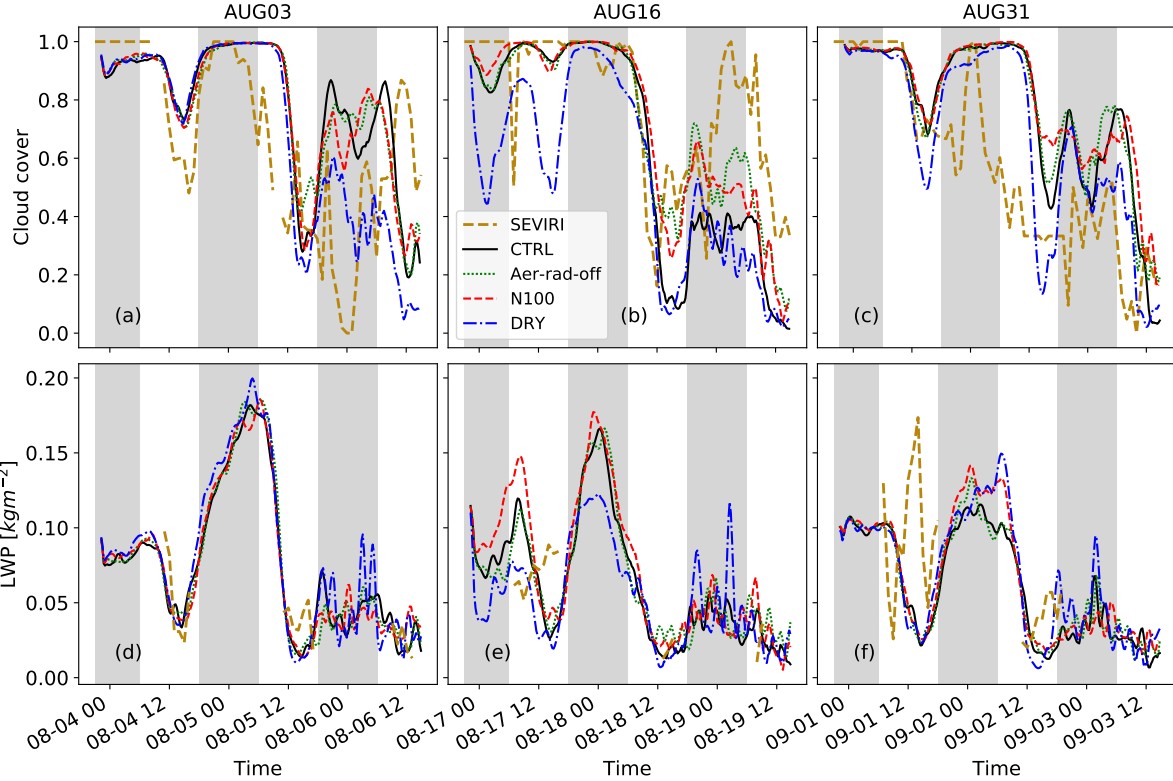

**Figure 5.** (a,b,c) Cloud cover and (d,e,f) liquid water path along the three trajectories as a function of time from SEVIRI (NASA) and the MIMICA simulations (liquid water path is in-cloud in MIMICA). The cloud cover and liquid water path output from MIMICA have a time resolution of 15 minutes and values are smoothed using a 1-hour moving average.

the absorbing aerosol that amplifies the reduction in cloud cover in CTRL (including aerosol-radiation interactions) compared to Aer-rad-off. Figure 6b shows that the LW cooling at the top of the MBL is lower in CTRL compared to Aer-rad-off during the second day, which contributes to a decrease in the MBL TKE (Figure 6e). Furthermore, the absorption of solar radiation

produces a stronger SW heating just below the cloud (around 2 $K \cdot day^{-1}$ difference between CTRL and Aer-rad-off, Figure 11e) compared to close to the surface (around 1 $K \cdot day^{-1}$ difference between CTRL and Aer-rad-off). This heating profile stabilizes the MBL in CTRL which also favors reduced buoyancy production. The net result is a clear reduction in cloud cover in CTRL compared to Aer-rad-off (Figure 5e). The entrainment rate is also substantially lower in CTRL compared to Aer-rad-off (Figure 6h) and actually reaches negative values during the second day of simulation, in agreement with a reduction of the

inversion height with time (Figure 6k). The above processes cause the MBL to be around 200 meters shallower in CTRL than in Aer-rad-off during the last day of simulation. There is a negligible difference in drizzle production between Aer-rad-off and CTRL, since the large number of aerosols in the MBL effectively suppresses precipitation (Figure 7h). As in AUG03, the CRE




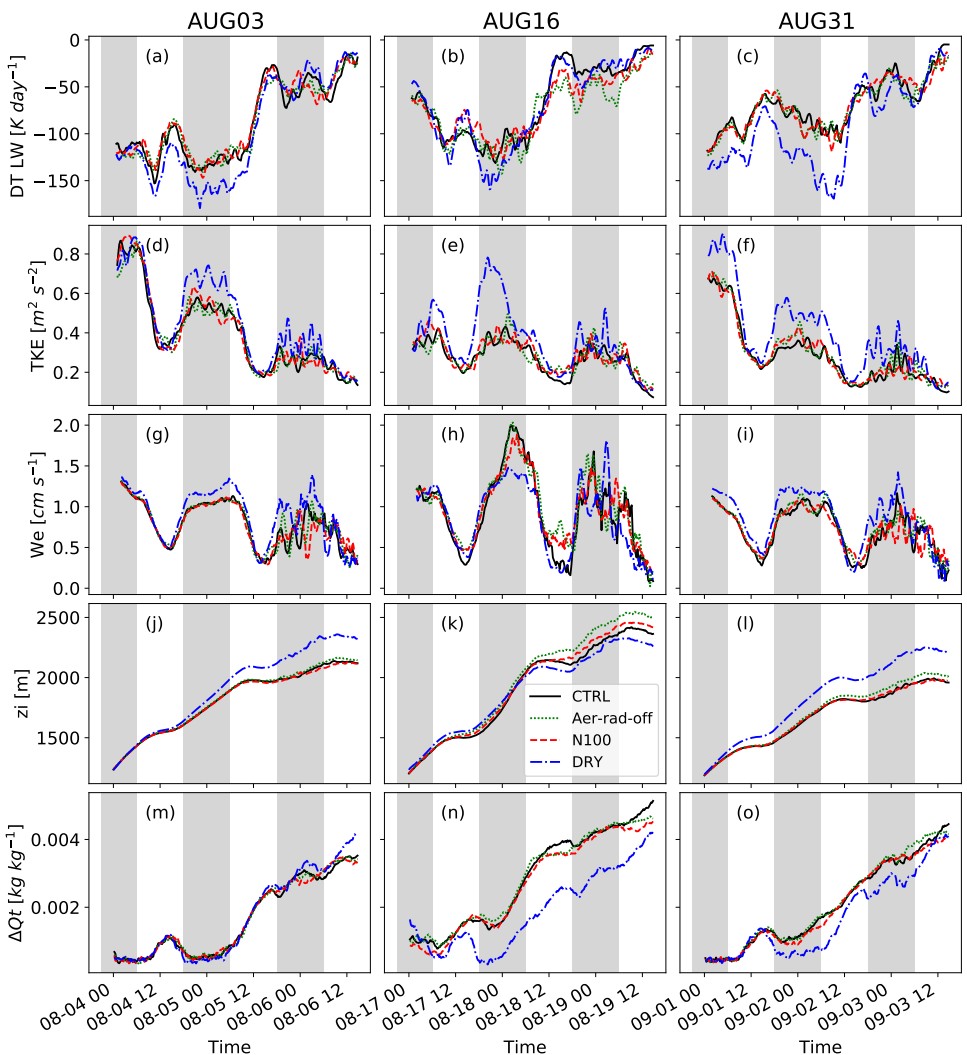

**Figure 6.** Temporal evolution of the simulated (MIMICA) domain-averaged: (a-c) LW cooling rate (DT LW) at the MBL top (corresponds to the maximum horizontal domain-averaged LW cooling rate), (d-f) MBL TKE, (g-i) Cloud-top entrainment rate (We), (j-l) Inversion height (zi), (m-o) Decoupling parameter ($\delta Q_t$). All output variables have 15 minutes temporal resolution. DT LW and TKE values are smoothed using a 1 hour moving average. The entrainment rate is smoothed with 2 hours moving average.

in AUG16 is less negative in CTRL than in Aer-rad-off (Table 2), but the difference between the simulations is larger due to the larger amounts of BBA in the AUG16 case.

For AUG31, the impact of the direct and semi-direct aerosol effects within the MBL are weaker than in AUG16, which is expected since the MBL is less polluted and the BBA plume remains mainly above the cloud (Figure 2). The mean SW heating difference between CTRL and Aer-rad-off is largest during the third day, reaching a maximum value of around $1 \ K \cdot day^{-1}$



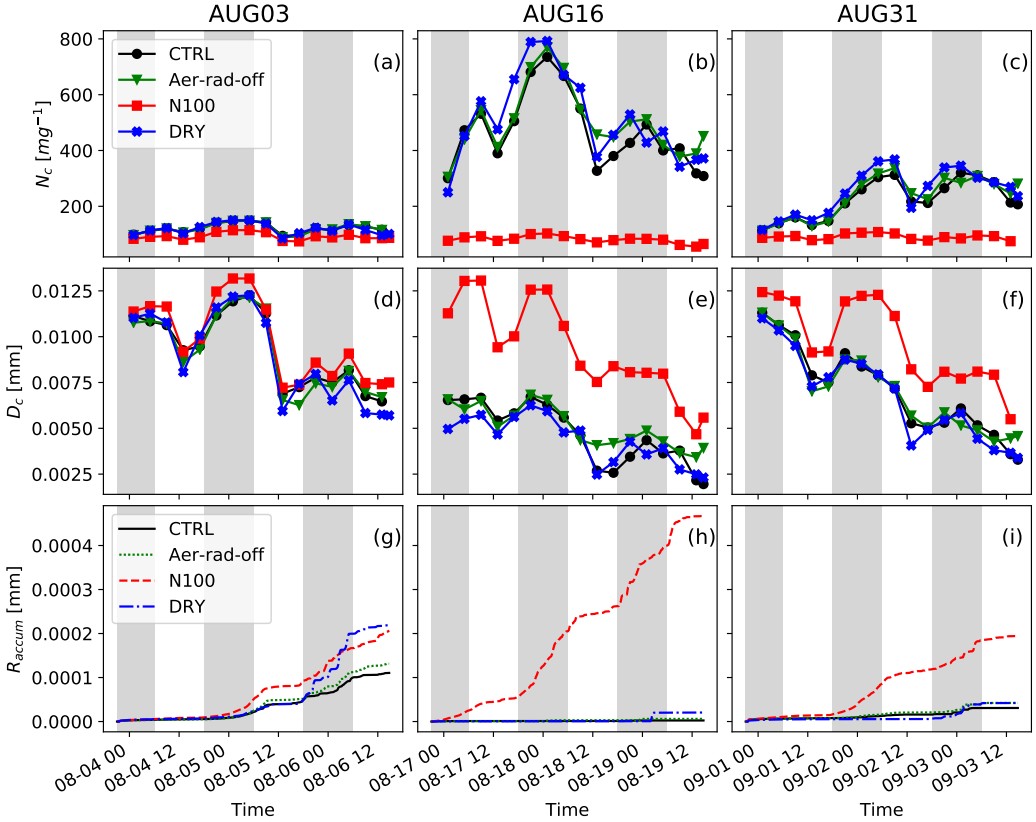

**Figure 7.** Temporal evolution of the simulated (MIMICA) domain-averaged (a-c) cloud droplet number concentration ($N_c$), (d-f) cloud droplet mean size ($D_c$), (g-i) accumulated precipitation ($R_{accum}$). The output of $N_c$ and $D_c$ have a 4 hours time resolution while the temporal resolution for the $R_{accum}$ output is 15 minutes.

just below the cloud (Figure 11i). Therefore, the differences between CTRL and Aer-rad-off regarding the cloud variables are smaller than in AUG16 (Figure 5). Similar to AUG03 and AUG16, the CRE is less negative in CTRL than in Aer-rad-off (Table 315    2).

To summarize, all three cases show clear differences in cloud cover due to direct and semi-direct effects. These differences occur regardless of the level of pollution, although they are more pronounced when pollution in high, and during broken cloud conditions (during the second and third day of simulation) compared to overcast conditions (first day of simulation). In other words, the efficiency of the direct and semi-direct aerosol effects appears to increase with broken cloud conditions. For 320    the AUG16 case, the SW heating during daytime is clearly larger within and below cloud in CTRL compared to Aer-rad-off (Figure 11). For the other two cases, the differences in mean daytime SW heating within and below clouds are less pronounced.



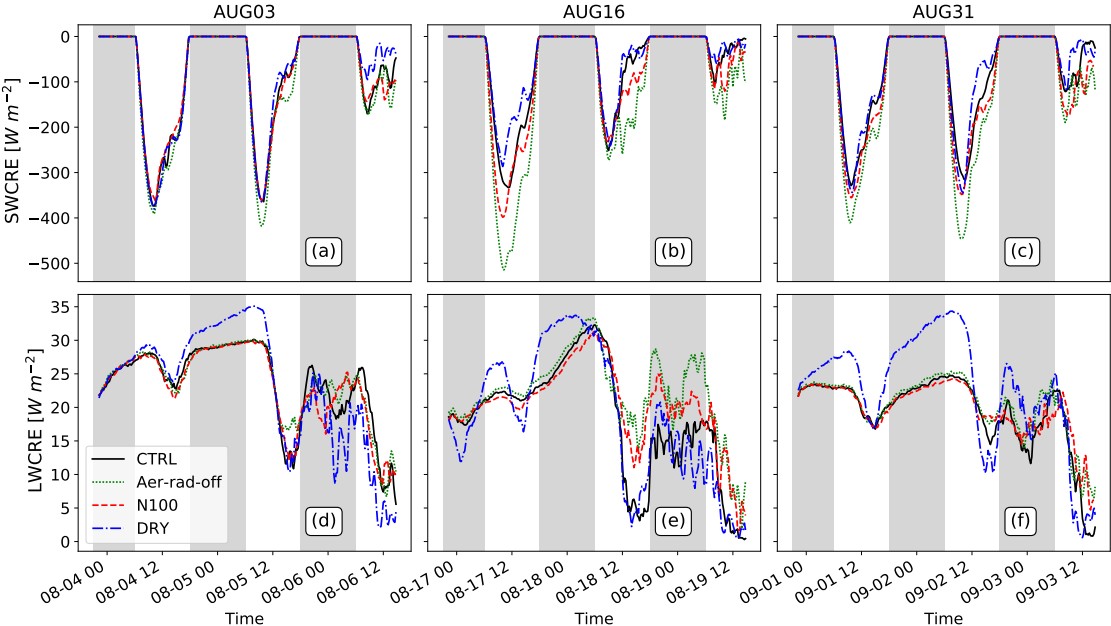

**Figure 8.** Simulated (MIMICA) time evolution of domain-averaged shortwave (a-c) and longwave (d-f) cloud radiative effects (SWCRE and LWCRE respectively) for AUG01, AUG03 and AUG31.

**Table 2.** Simulated (MIMICA) domain-averaged diurnal SW and LW cloud radiative effects (in $Wm^{-2}$).

| DAYS | CTRL | | | Aer-rad-off | | | N100 | | | DRY | | |
|---|---|---|---|---|---|---|---|---|---|---|---|---|
| **AUG03** | **SW** | **LW** | **SW+LW** | **SW** | **LW** | **SW+LW** | **SW** | **LW** | **SW+LW** | **SW** | **LW** | **SW+LW** |
| **1** | -219.2 | 26.1 | -193.0 | -238.0 | 25.8 | -212.0 | -209.9 | 25.4 | -184.0 | -224.9 | 27.5 | -197.0 |
| **2** | -163.4 | 22.1 | -141.0 | -195.9 | 23.8 | -172.0 | -168.9 | 22.5 | -146.0 | -154.0 | 24.3 | -130.0 |
| **3** | -95.6 | 15.6 | -80.0 | -115.2 | 14.8 | -100.0 | -104.4 | 14.3 | -90.0 | -45.2 | 8.9 | -36.0 |
| **AUG16** | **CTRL** | | | **Aer-rad-off** | | | **N100** | | | **DRY** | | |
| **1** | -192.2 | 21.9 | -170.0 | -313.2 | 22.6 | -291.0 | -233.7 | 21.1 | -213.0 | -149.3 | 23.4 | -126.0 |
| **2** | -99.6 | 15.6 | -84.0 | -169.1 | 21.7 | 147.0 | -124.2 | 21.1 | -103.0 | -89.2 | 15.5 | -74.0 |
| **3** | -36.2 | 6.7 | -30.0 | -94.8 | 12.2 | -83.0 | -60.8 | 10.5 | -50.5 | -27.2 | 4.9 | -22.0 |
| **AUG31** | **CTRL** | | | **Aer-rad-off** | | | **N100** | | | **DRY** | | |
| **1** | -181.6 | 20.5 | -161.0 | -240.0 | 20.6 | -219.0 | -209.4 | 20.2 | -189.0 | 178.0 | 23.8 | -154.0 |
| **2** | -160.9 | 21.0 | -140.0 | -242.4 | 21.9 | -220.0 | -189.1 | 21.2 | -168.0 | -140.2 | 23.7 | -116.0 |
| **3** | -62.0 | 10.4 | -52.0 | -112.8 | 12.8 | -100.0 | -104.5 | 13.9 | -91.0 | -42.6 | 9.1 | -34.0 |





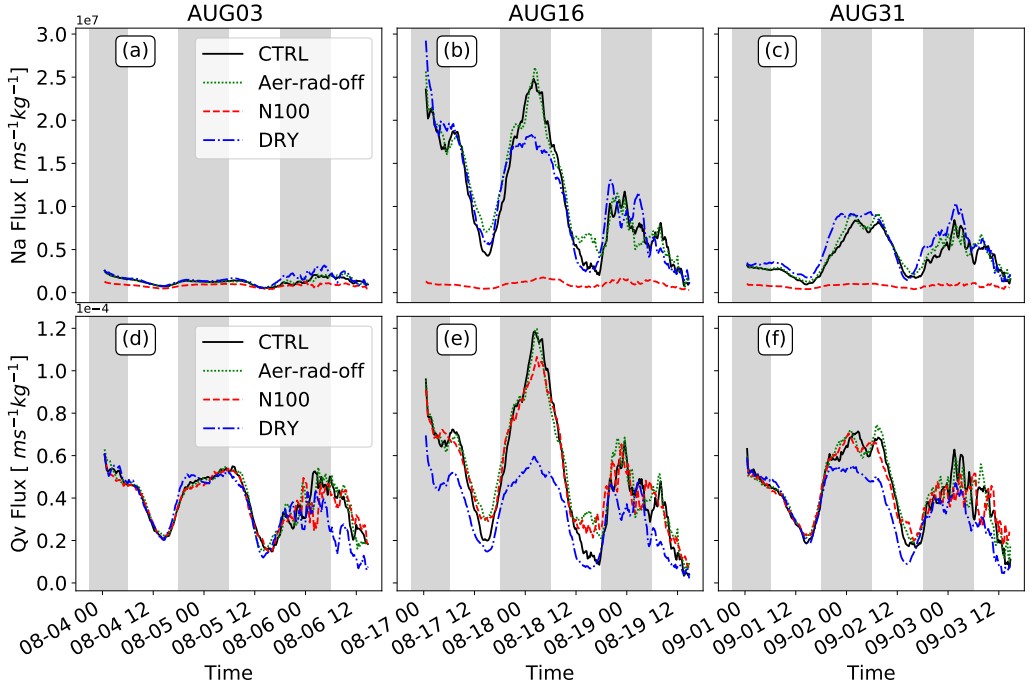

**Figure 9.** Temporal evolution of the simulated (MIMICA) domain-averaged aerosol ($Na$) and water vapor mixing ratio ($Q_v$) fluxes at the top of the MBL for AUG03, AUG16 and AUG31

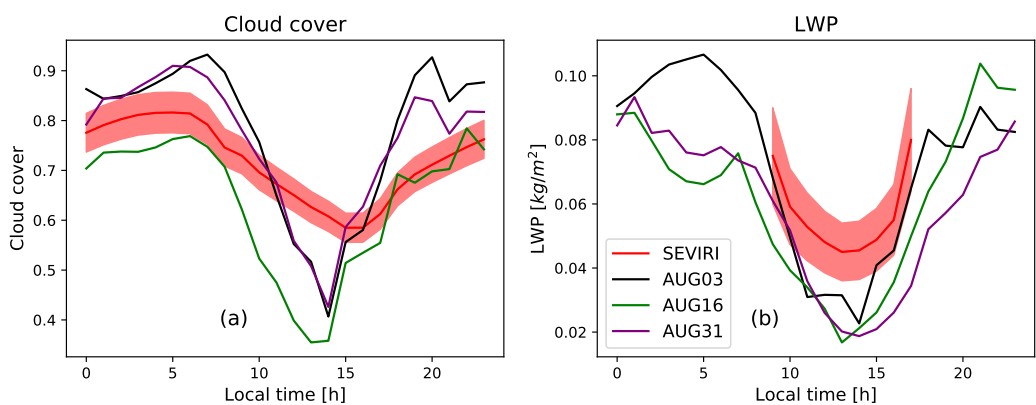

**Figure 10.** Average diel cycle of (a) cloud cover and (b) liquid water path from SEVIRI (CLAAS-3) and MIMICA. Values from SEVIRI (CLAAS-3) correspond to 10 years of data during August over the geographical area covered by the trajectories. The retrievals of LWP are based on the shortwave channels, hence they are available only during the sunlit hours. For MIMICA, time-average values along each trajectory are shown.





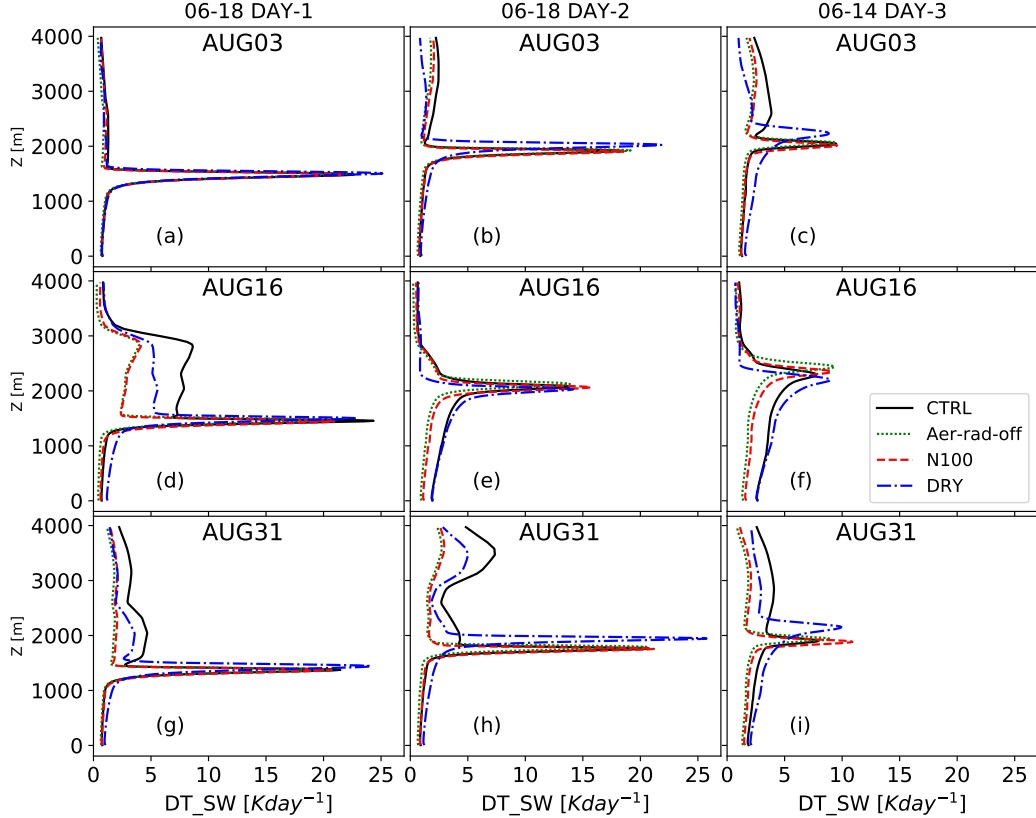

**Figure 11.** Simulated (MIMICA) mean daytime (between 6 and 18 h) profiles of SW heating for the three days for each case and each sensitivity experiment.

### 3.4 Influence of aerosol indirect effects

In order to obtain an individual estimate of the aerosol indirect effect, we compare the N100 and Aer-rad-off simulations. Note that we cannot compare N100 with CTRL as this would show the combined direct, semi-direct, and indirect aerosol effects. The low $N_a$ in N100 produces mean daytime SW heating profiles in the FT similar to those in Aer-rad-off (Figure 11), showing that the impact of aerosols on radiation is small in N100, and that it is reasonable to compare Aer-rad-off and N100 to derive an estimate of the aerosol indirect effect.

In general, the indirect aerosol effect should primarily be visible in the cloud microphysical parameters. In all Aer-rad-off simulations, the $N_c$ is also higher than in N100 (Figure 7a-c). This increase leads to a reduction in the average cloud droplet radius (Figure 7d-f) and an inhibition of drizzle (Figure 7g-i). The largest impact on the drizzle production occurs for AUG16, most likely as a consequence of the relatively high humidity content of the BBA plume for this case. The precipitation sensitivity to $N_a$ is most pronounced during nighttime, when most of the drizzle is produced.





The difference in cloud cover between N100 and Aer-rad-off (Figure 5) varies with time and is not consistent for all cases, in particular during daytime. During nighttime, and especially towards the end of the simulation, there is a tendency for N100
to show lower cloud cover values than Aer-rad-off, i.e. that higher aerosol concentrations lead to more clouds - consistent with the lower precipitation values and the aerosol indirect effect. However, due to the small and inconsistent differences between N100 and Aer-rad-off, it is difficult to make a robust assessment of the indirect effect of BBA plumes on cloud evolution. Furthermore, for AUG16, the cloud cover differences between Aer-rad-off and N100 during and after the second day of simulation are smaller than between each of those experiments and CTRL. This suggests that, for this situation, the
aerosol heating effect (semi-direct effect) is more important than the indirect aerosol effect in terms of affecting the Sc cloud properties and the transition into cumulus.

The increase in $N_a$ between N100 and Aer-rad-off produces a higher (in absolute terms) SW CRE for all cases (Table 2). There are two reasons: first, the higher $N_c$ and smaller droplet radii increase the cloud albedo; second, aerosols are transparent to radiation in Aer-rad-off, producing darker clear skies and brighter cloudy skies. In contrast, the SW CRE generally decreases
when comparing CTRL with N100, i.e. when both direct and indirect effects are considered. The main reasons for this difference are the clear-sky albedo which is higher in CTRL than in N100 due to more aerosols as well as that the cloud cover tends to be smaller in CTRL than in N100 during daytime.

### 3.5 Influence of moisture in the BBA plume

The impact of moisture within the BBA plume on cloud evolution is examined by comparing CTRL with DRY. Figure 5 shows
that there is a clear difference in cloud cover between all CTRL and DRY cases, but the impact differs between AUG16 and the other two cases.

In AUG03 and AUG31, there are enhanced levels of moisture above and in contact with the MBL, in particular during the second half of the simulation (AUG03) or after the first day of simulation (AUG31, Figure 3). This additional moisture could favor the high values of cloud cover observed in CTRL compared to DRY (Figure 5). However, differences in moisture
entrainment between CTRL and DRY are generally small, in particular for AUG03 (Figure 9d). The most likely reason for the higher cloud cover in CTRL compared to DRY is instead that the MBL is shallower in the former (Figure 6j-l). Enhanced levels of moisture above the MBL can reduce the net LW cooling at cloud top (see Section 1). Figure 6a and 6c also show that before midday of the second day, the LW cooling at the MBL top is generally lower in CTRL compared to DRY in AUG03 and AGU31. In CTRL, the reduced LW cooling leads to a reduction of the MBL TKE and cloud-top entrainment, which reduces
the MBL growth compared to DRY (Figure 6). More clouds in CTRL than in DRY results in a more negative daytime CRE, in particular during the last two days of simulation.

The effect of moisture on cloud evolution is more complex for the AUG16 case. During the first 1.5 days of simulation, moisture is clearly enhanced in the FT in CTRL compared to DRY (Figure 3) meaning that moister air is also progressively entrained into the MBL (Figure 9d-f). Similar to the AUG03 and AUG31 cases, the additional moisture contributes to higher
cloud cover and LWP values in CTRL compared to DRY for AUG16 (Figure 5). Since the cloud cover and LWP in DRY are substantially lower than in CTRL during the first night and day, the domain average LW cooling at the top of the MBL





is reduced (Figure 6b). Therefore, unlike in AUG03 and AUG31, the domain averaged LW cooling at the MBL top is not consistently higher in DRY than in CTRL for the AUG16 case during the first 1.5 days of simulation (Figure 6b). Between the second midnight and the following morning of AUG16, most of the humid BBA plume has entrained into the MBL in the
CTRL simulation. In CTRL, moist entrainment maintains the Sc cloud deck during this period, while the progressive reduction in humidity above the MBL facilitates MBL deepening. In contrast, DRY experiences an early (before sunrise) cloud break up with an associated reduction in the domain averaged MBL top LW cooling, MBL TKE, and entrainment rates with respect to CTRL, which leads to a slowing of the MBL growth (Figure 6k). During and after day 2 of AUG16, The MBL in CTRL remains moister than DRY, particularly in the lower half of the MBL (Figure 3). During this period, the differences in cloud
cover between CTRL and DRY are not consistent, especially between midday of day 2 and the third midnight. The stronger MBL decoupling in CTRL compared to DRY prevents moisture fluxes from the lower MBL to reach the cloud base in CTRL, which to some extent limits cloud growth in CTRL with respect to DRY.

### 3.6 Quantification of the radiative effects of aerosols and moisture

We compare the overall radiative impact of aerosols due to the direct and semi-direct effects, due to the indirect effect and
due to all aerosol effects. Table 3 shows that the time-mean domain-averaged radiative effect at the top of the model domain is dominated by SW radiative effects. In general, positive values in the SW (i.e. a warming) can be caused by absorption of aerosols above clouds (which reduces the upwelling SW radiation at the top of the model domain) or by a reduction in cloudiness or cloud albedo. In contrast, the SW effect is negative if the cloud albedo or cloudiness increases or if aerosols are present under clear-sky conditions. In the LW, a positive effect (warming) can be caused by an increase in cloudiness or
moisture.

The mean direct and semi-direct aerosol effect is positive for AUG16 and AUG31 but negative for AUG03. Table 4 show that the SW radiative effect is positive during the first two days in the three cases, with higher values in AUG16 and AUG31 as a result of the presence of absorbing aerosols over the Sc cloud deck and the reduction in cloudiness in CTRL compared to Aer-rad-off. However, during the third day, the radiative effect becomes negative as the increase in clear-sky albedo (in CTRL
compared to Aer-rad-off) becomes larger than the decrease in albedo due to fewer clouds. The mean indirect aerosol effect is negative for all three cases (AUG03, AUG16, and AUG31). This result is expected as there was no substantial impact on cloud cover and because the smaller cloud droplets in CTRL compared to N100 increased the cloud albedo. The total radiative effect from the combination of direct, semi-direct and indirect aerosol effects is also negative for all cases. This shows that for these three cases, and averaged over the whole simulations time, the net effect of the biomass burning aerosols is to cool the system,
with the indirect effect dominating over the direct and semi-direct aerosol effects.

The enhanced moisture associated with the BBA plume also leads to a negative mean radiative effect (cooling) for all three cases (Table 3). The main reason is that the enhanced moisture helps to sustain the Sc cloud deck which increases the albedo of the system. Note that for all three cases, the cooling caused by the enhanced moisture is about as large as the sum of all the aerosol effects (direct + semi-direct + indirect).



**Table 3.** Domain-averaged and time-averaged radiative effect (in $Wm^{-2}$) due to aerosol direct and semi-direct effects, indirect aerosol effects, all aerosol effects combined, and enhanced moisture. The mean values are calculated over 64 hours of MIMICA simulations.

| | Direct and semi-direct effect CTRL - Aer-rad-off | | | Indirect effect Aer-rad-off - N100 | | | All aerosol effects CTRL - N100 | | | Moisture effect CTRL - DRY | | |
|---|---|---|---|---|---|---|---|---|---|---|---|---|
| Case | SW | LW | SW+LW | SW | LW | SW+LW | SW | LW | SW+LW | SW | LW | SW+LW |
| AUG03 | -0.8 | -0.3 | -1.1 | -3.7 | 0.6 | -3.1 | -4.5 | 0.4 | -4.1 | -7.4 | 2.7 | -4.7 |
| AUG16 | 12.6 | -4.4 | 8.2 | -19.2 | 2.0 | -17.2 | -6.6 | -2.4 | -9.0 | -10.1 | -0.4 | -10.5 |
| AUG31 | 3.4 | -1.0 | 2.4 | -8.4 | 0.7 | -7.7 | -5.0 | -0.3 | -5.3 | -6.7 | 0.9 | -5.8 |

**Table 4.** Mean daytime radiative effect (in $Wm^{-2}$) at the top of the model domain in AUG03, AUG16 and AUG31 due to aerosol direct an semi-direct effects, indirect aerosol effects, all aerosol effects combined and enhanced moisture.

| AUG03 | Direct and semi-direct effect CTRL - Aer-rad-off | | | Indirect effect Aer-rad-off - N100 | | | All aerosol effects CTRL - N100 | | | Moisture increase CTRL - DRY | | |
|---|---|---|---|---|---|---|---|---|---|---|---|---|
| DAYS | SW | LW | SW+LW | SW | LW | SW+LW | SW | LW | SW+LW | SW | LW | SW+LW |
| 1 | 0.8 | 0.3 | 1.1 | -13.9 | 0.3 | -13.6 | -13.1 | 0.7 | -12.4 | 5.8 | 0.9 | 6.7 |
| 2 | 7.9 | -1.6 | 6.3 | -11.8 | 1.3 | -10.5 | -3.9 | -0.3 | -4.2 | -9.1 | 0.4 | -8.7 |
| 3 | -17.8 | 0.8 | -17.0 | 7.1 | 0.6 | 7.7 | -10.7 | 1.4 | -9.3 | -51.2 | 10.2 | -41.0 |
| **AUG16** | **Direct and semi-direct effect CTRL - Aer-rad-off** | | | **Indirect effect Aer-rad-off - N100** | | | **All aerosol effects CTRL - N100** | | | **Moisture increase CTRL - DRY** | | |
| DAYS | SW | LW | SW+LW | SW | LW | SW+LW | SW | LW | SW+LW | SW | LW | SW+LW |
| 1 | 51.2 | -0.7 | 50.5 | -65.1 | 1.5 | -63.6 | -14.0 | 0.8 | -13.2 | -41.2 | -1.0 | -42.2 |
| 2 | 17.9 | -6.8 | 11.1 | -29.2 | 1.1 | -28.1 | -11.3 | -5.7 | -17.0 | -9.1 | 0.1 | -9.0 |
| 3 | 0.3 | -6.8 | -6.5 | -15.6 | 2.3 | -13.3 | -15.3 | -4.4 | -19.7 | -7.5 | 2.2 | -5.3 |
| **AUG31** | **Direct and semi-direct effect CTRL - Aer-rad-off** | | | **Indirect effect Aer-rad-off - N100** | | | **All aerosol effects CTRL - N100** | | | **Moisture increase CTRL - DRY** | | |
| DAYS | SW | LW | SW+LW | SW | LW | SW+LW | SW | LW | SW+LW | SW | LW | SW+LW |
| 1 | 9.6 | -0.1 | 9.5 | -15.8 | 0.4 | -15.4 | -6.2 | 0.3 | -5.9 | -3.0 | 0.4 | -2.6 |
| 2 | 19.1 | -0.9 | 18.2 | -37.5 | 0.7 | -36.8 | -18.4 | -0.2 | -18.6 | -20.1 | 1.9 | -18.2 |
| 3 | -14.2 | -2.4 | -16.6 | 10.3 | -0.9 | 9.4 | -3.9 | -3.3 | -7.2 | -19.1 | 6.6 | -12.5 |

## 4 Discussion and conclusions

In this study, we have used large-eddy simulation in a Lagrangian setup to explore how the stratocumulus cloud cover, boundary layer evolution, and stratocumulus-to-cumulus transitions over the Southeast Atlantic are affected by the individual impacts of absorbing aerosols and moisture within biomass burning plumes. We initialized and forced our model with meteorological conditions corresponding to three different periods during August 2017 (AUG03, AUG16 and AUG31). These situations were



clearly different with respect to the levels of pollution and water vapor in the plume, and also regarding when the plume appeared in the free troposphere and when it mixed into the cloud layer and MBL. The selection of three cases can be useful to generalize some of the impacts of the BBA layers on the Sc clouds and SCTs. The analysis and comparison of multiple situations also helps giving a wider perspective on the topic, since there is only one previous study that investigates the effects of biomass burning aerosols on SCTs using large-eddy simulation with meteorological conditions specific for the Southeast

Atlantic (Diamond et al., 2022).

  An evaluation of the model results against satellite retrievals showed that the model reproduced the diel cycle of cloud cover and liquid water path reasonably well from a climatological perspective. For all three periods examined, the simulations showed SCTs that were broadly consistent with the theory of a "deepening-warming" transition (Bretherton and Wyant, 1997), which is in agreement with results from previous studies under relatively high aerosol concentration conditions [e.g. Yamaguchi et al.

(2015); Zhou et al. (2017); Diamond et al. (2022)]. The drizzle increased slightly when aerosol concentrations were reduced and there was a clear decrease in the cloud droplet number concentration. However, none of the experiments showed signs of a relevant impact of drizzle on the SCT, not even when the aerosol concentrations were lowered to background levels. The reason is likely that there was little drizzle formation in all cases and that experiments with an even lower aerosol concentration are needed to explore a potential "drizzle-driven" transition in MIMICA [e.g. Yamaguchi et al. (2015, 2017); Erfani et al.

(2022); Diamond et al. (2022)]. These findings further amplify the existing interest in exploring the SCT in different LES models within a common framework. In this context, the Southeastern Atlantic Stratocumulus Transitions with Aerosol-Rain-Radiation interactions (SEA STARR) large eddy simulation intercomparison project is currently being conducted in order to investigate the SCT in several LES models (including MIMICA) under same meteorological conditions and aerosol forcings, and improve our comprehension of the factors responsible for the differences in the SCT between models.

The semi-direct effect of absorbing aerosols that were in contact with, or mixed within, the MBL was found to be substantial, especially in highly polluted situations, and in particular during daytime and during broken cloud conditions. Our simulation results imply that biomass burning aerosols have the potential to speed up SCTs through local semi-direct effects. However, the influence will be dependent on the state of the SCT as well as the time of the day when the absorbing aerosol makes contact with the cloud deck. In our simulations, the forcing conditions were the same in all experiments, with impacts of the biomass

plume on temperature and winds always included in the free troposphere. Therefore, we were not able to explore any free tropospheric semi-direct aerosol effects on cloud cover (as in e.g. Yamaguchi et al. (2015), Zhou et al. (2017) and Diamond et al. (2022)), or the impacts of large-scale subsidence changes caused by aerosol heating (as in Diamond et al. (2022)). Thus, we cannot say if semi-direct effects of aerosols in the free troposphere can reduce the relative importance of local semi-direct effects in the MBL as shown in Yamaguchi et al. (2015), Zhou et al. (2017) and Diamond et al. (2022).

The moisture associated with the aerosol plume had a clear impact on the MBL and SCT for the three periods analyzed. When located mostly above the Sc deck, the LW radiative effect of the humidity slowed down the MBL deepening, in particular during the night. These results are consistent with those obtained by Yamaguchi et al. (2015) and Zhou et al. (2017). However, in contrast with Yamaguchi et al. (2015) and Zhou et al. (2017), we did not find that the LW radiative effect of the water vapor above the MBL cause cloud breakup. The reason for this difference is that the enhanced moisture in the FT was never clearly





separated from the cloud in our simulations. Thus, the moisture from the plume could always entrain the cloud and the MBL,
which led to an increase in cloud cover and a delay of the SCT, in agreement with Yamaguchi et al. (2015).

We note that the conclusions drawn regarding the plume impact on the SCT may depend on the definition of the SCT, and
that there are different ways to define the SCT in the literature. Here we have used the definition employed by Sandu and
Stevens (2011) and Zhou et al. (2017) (see section 2.2) based on a cloud cover threshold of 50%. Recently, Erfani et al. (2022)
used a similar approach, but they also considered that the cloud cover should remain below 50%, either during the following
24 hours of simulation or until the simulation end. If we use the same approach as Erfani et al. (2022), the SCT happens one
day later in several of our simulations (including two of the control simulations: AUG03 and AUG31). Yamaguchi et al. (2015)
investigated the decrease in the domain mean albedo at the beginning and end of their simulations to obtain a measure of the
amplitude of the SCT, and thereby also an estimate of the pace of the SCT, following Sandu and Stevens (2011). This criterion
is not directly comparable with the ones based on cloud cover. A common metric among different studies would be useful to
make comparisons more robust. Since all the mentioned criteria regarding the SCT have their flaws and do not fully capture
the complexity of the phenomena, we have avoided to rigorously use the 50% threshold in cloud cover to conclude that the
semi-direct effect favours while the moisture delays the SCT. Instead, we have looked at the differences in the evolution of the
cloud cover between the experiments and used the 50% threshold as a "soft" reference of the SCT.

In our simulations, the indirect aerosol radiative effect always dominated over the direct and semi-direct radiative effects,
and the absorbing aerosol plume produced an average net radiative cooling effect over the three days of simulation of about -4
to -9 $Wm^{-2}$. These results are consistent with Lu et al. (2018), who used a regional model to investigate the effects of biomass
burning aerosols during two months over the SEA. However, our estimate can be influenced by the fact that our experiments
did not include any aerosol effects in the FT and that our BBA plumes were not clearly separated from the cloud deck. For
instance, the semi-direct effect of absorbing aerosols located above and separated from the cloud deck might contribute to
additional cooling as in Che et al. (2020). Our simulations also showed that the moisture accompanying the absorbing aerosol
in the biomass burning plume produced an additional cooling effect that was as about as large as the total aerosol radiative
effect itself.

*Data availability.* Modelling data sets used in this study are available at https://bolin.su.se/data/baro-perez-2023-biomass-burning-1 (last
access: 12 September 2023)



*Author contributions.* ABP and MSD designed the experiments. MSD provided the methodology to force MIMICA. ABP implemented the aerosol-radiation interactions in the model with imputs from JT and HK, and under the supervision of AMLE, JS and MS. ABP performed the simulations, provided all visualisations and wrote the manuscript with inputs and revisions from all the coauthors. AMLE, FAMB and AD revised the manuscript and participated in the analysis of results. AD provided the SEVIRI (CLAAS-3) dataset. DP and HL provided, respectively, the SEVIRI (NASA) dataset and the Lagrangian trajectories used to force the model.


*Competing interests.* The authors declare that they have no conflict of interest.

*Acknowledgements.* This study was funded by the Swedish National Space Agency grant 16317. Additional funding was provided by the European Commission, H2020 Research Infrastructures (FORCeS, grant no. 821205) and the Swedish Research Council (2020-04158). The computations and data handling were enabled by resources provided by the Swedish National Infrastructure for Computing (SNIC) at the National Supercomputer Centre (NSC) partially funded by the Swedish Research Council through grant agreement no. 2018-05973. We are grateful for the technical assistance provided by Matthias Brakebush (SU) and Hamish Struthers (NSC) in running the MIMICA LES model on the NSC resources. Many thanks to Jens Redemann for his valuable comments on the manuscript.




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
