# Peer review of "Comparing the simulated influence of biomass burning plumes on low-level clouds over the southeastern Atlantic under varying smoke conditions."

_EGUsphere, 2023_

## Author Response (AR1)

We want to thank the reviewers for their constructive feedback. Below you will find our response detailing the modifications we have made in relation to the comments and suggestions.

**Reviewer 1**

- Starting on Line 35, this explanation of the deepening-warming mechanism during SCT seems quite unclear to me: (1) it sounds strange to say "latent heat fluxes ... are positively buoyant", (2) it is not entirely true that "...the entrained air is positively buoyant...". There are quite a lot of papers on the possibility of "buoyancy reversal" when dry FT air mixes with cloudy air near the cloud top …

The sentences with expressions 1 and 2 highlighted by the reviewer were modified in the following way:

1- "Warmer SSTs lead to increased surface latent heat fluxes that increase the turbulent kinetic energy (TKE) and enhance turbulence in the boundary layer." (we removed "positively buoyant")

2- "If the mixture between the warm entrained air and the cloudy air is positively buoyant, the mixing between the cloud and the sub-cloud layers is reduced." (here we exclude the "buoyancy reversal" cases)

- Line 91: "This delay ...", I don't know what "delay" this is …

We should have written "effect" instead of "delay". It has been corrected.

- Line 106: Are you emphasizing some kind of difference between "diurnal" and "diel"?

Yes. "Diurnal" sometimes can be interpreted as only during the daylight hours. Therefore we have used the term "diel" which refers to a 24 hour period regardless of day or night.

- Line 123: "However ...": I am not an expert on this. What if there are more than one aerosol in one droplet of rain due to collision etc.?

For more than one aerosol in one droplet, the model does a simple (but not completely unrealistic) assumption that the two aerosols merge into one upon evaporation. If we assume that the aerosols are water soluble, then this is quite likely. We have added the following sentence after Line 123:

"When collision-coalescence processes occur, the model does a simple assumption that the aerosols from the involved droplets merge into one upon evaporation."

- Line 141: This divergence is the same for all cases, all heights and all times?

Yes, this clarification was added to the sentence.

- Line 330: "The largest impact on the drizzle production ...": This case also just has the most aerosols in the PBL. Isn't that a stronger impact than humidity?

Yes, we agree with the reviewer and we have changed the sentence accordingly:

"The largest impact on the drizzle production occurs for AUG16, as a consequence of the largest Na difference between Aer-rad-off and N100."

- Line 336: "However, due to ...": Please elaborate a little here on the difficulty as this is quite important …

The text was modified in the following way:

'"However, the impact of the indirect effect on cloud cover evolution is generally only seen if there is a substantial drizzle. This is not the case for our simulations, where the drizzle is small even in the N100 case (thus, the ability of MIMICA to simulate drizzle should be evaluated further). In consequence, there are small and inconsistent differences between N100 and Aer-rad-off that prevent us from making a robust assessment of the indirect effect of BBA plumes on cloud evolution."

- I feel like towards the end of the paper the impact of the direct and semi-direct effects is toned down a little ... Why? I think this is an important thing I learned from the paper as a non-expert on aerosols.

In our conclusions (Line 425) we highlight that there is a substantial semi-direct aerosol effect in situations of contact under highly polluted conditions. We don't consider that more emphasis on this is needed since other studies have found the semi-direct effect of absorbing aerosols in the free troposphere (which we cannot evaluate in our simulations) to dominate over the "in-cloud" semi-direct effects (the one we can evaluate). Since in our study we are missing an evaluation of the first effect mentioned above, we prefer not to overemphasize the impact of the "in-cloud" semi-direct effect, which might not be the most important in a more realistic simulation.

**Reviewer 2:**

Major comments:

- This may be beyond the scope of this paper, but the authors in a couple places effectively note noise in the results (e.g., L337 "due to the small and inconsistent differences [between two cases], it is difficult to make a robust assessment…"). Have you performed any sensitivity tests, e.g. slight perturbations in the initial conditions to try get a better understanding of the significance of the inter-scenario uncertainties?

We have not performed such sensitivity tests. However, there is an ongoing parallel study (an LES model intercomparison project) that uses a similar metodology to the one used in this work. In that study, MIMICA simulations are compared with simulations of other LES models for 3 different scenarios. This study will give a better perspective of the variability and uncertanties in the simulations of the participant LES models.

- Much of the discussion focuses on the differences between two (or more) different cases to isolate the different aerosol effects; in some cases it's rather difficult to see these interpretations (specifically Figs 8, 9, and 11). I wonder if it would be too busy to e.g. add a panel to Fig 11 to show the SW heating differences (from the discussion on L320) for each set of differences/effects, rather than e.g. trying to remember which colors are differenced to isolate just indirect effects. In a few places the wording on these differences could have been more straightforward, e.g. L285 "the decrease in cloud cover in CTRL compared to aer-rad-off" as written isn't clear whether you mean to highlight a decrease in cloud fraction (or altitude?) over time (differently for the two scenarios) or that CTRL has less cloud than aer-rad-off on all three days. Another instance is L344 "SW CRE generally decreases when comparing CTRL with N100"—not very clear whether this refers to changes over time or

between the two. Alternatively, plotting the differences in Fig 8 might help to clarify the meaning, but runs the risk of being too busy.

Calculating and plotting SW heating differences from the profiles in Figure 11 can be confusing and missleading because the aerosol and cloud layers are not exactly at the same altitude in the different simulations (CTRL, Aer-rad-off, N100 and DRY) . It can therefore be that such plots show the difference in SW heating rates between, for instance, the cloud in one simulation and the aerosol plume in another simulation.

We decided to remove Figure 8 since it was not a key figure in the description of the results. There is now a new Figure 8 (stacked bar plot showing the information given previously in Table 2) which is adequate and sufficient to understand what is explained in the text.

L285 was rewritten in the following way:

"… a combination of two factors: the smaller cloud fraction in CTRL during daytime compared to Aer-rad-off (especially during the second day in the three cases), which reduces the albedo, …"

L344 was rewritten in the following way:

"SW CRE is in general smaller in CTRL than in N100"

Additionally, L342 was modified to be more coherent:

"The highest Na values in Aer-rad-off compared to N100 produce the highest (in absolute values) SW CRE  in all Aer-rad-off cases."

- Relatedly, in the discussion of quantifying the distinct effects (Table 4), I found myself wishing for a figure to better digest the findings. Perhaps some key differenced parameters from one of the earlier figures, or this table converted into stacked bar plots to show the relative radiative effects relative to one another? Another option would be shaded backgrounds in Table 4, if that's a format allowed in ACP. Just a thought that might help to better illustrate the results.

We thank the reviewer for the suggestion and have replaced Tables 2, 3 and 4 with the new Figures 8, 12 and 13 respectively. The figures show the same information displayed before in the tables, but using stacked bar plots.

- The SEVIRI cloud values (especially cloud fraction) deviate significantly from the simulated conditions (e.g. Fig 5). The authors mention that there are relatively "few values per time," and it's expected to have satellite cloud retrieval gaps over the SEA, but a better sense of how much data are actually going into these figures might be helpful. Relatedly, are there specific criteria for what trajectories are considered "relatively close" for averaging purposes? (L255).

"Few values per time" : There are three (four in AUG16) values per time. Each value corresponds to one of the contiguous trajectories selected to calculate the average in each case (AUG03, AUG16, AUG31). This information is now included in the text.

We selected three contiguos trajectories (four in the case of AUG16) from SEVIRI, that are located at a distance less than 285 km from each of the simulated trajectories. We added this information to the text.

- It would be useful to cite Cochrane et al., 2022 (https://doi.org/10.5194/amt-15-61-2022) in the discussion section as well. Admittedly the framing is a bit different, but considering both are using data from August 2017 in the southeast Atlantic, it would be interesting to place the time-evolved heating rates presented here in the context of their observationally-constrained aerosol and water vapor heating. Specifically how one might reconcile the differences between this paper's Figure 11 and the more vertically-distributed heating rates (their Fig 7).

Figure 7 of Cochrane et al. (2022) shows, for the cases they study, the mean vertical profiles of heating rates between cloud top (1-2 km) and 10 km. In contrast, in our study, the profiles extend between the surface (boundary layer) and 4 km. To analyse the vertical distribution of the heating rates in a similar way as Cochrane et al.(2022) we would have to focus only on the part of our model domain that is in the free troposphere (FT). However, as mentioned in section 2.1, our model is nudged towards temperatures and horizontal winds provided by the GEOS-5 model in the free troposphere. Furthermore, as pointed out in section 3.3 (line 273), we note that we, for this specific reason, exclude above-cloud semi-direct effects from the comparison. This means that the FT heating rates shown in Fig 11 (in our manuscript) are not very relevant to our study because the forcing values from GEOS-5 mostly determine what happens with temperature and winds in the FT.

For the above reason, we think it is not appropriate to make a comparison with Cochrane et al. 2022. We think that it may be confusing to discuss a variable (heating rates above the FT) that is not actually modeled directly and analyzed in our study.

- Any idea what's happening with DRY's precip (Fig 7g-h) with that uptick on the last day (particularly AUG03)?

About the increase of precipitation on the last day of the DRY case (specifically in Fig 7g) we added the following paragraph in the section 3.4.

"Somewhat surprising, the AUG03 DRY simulation displays an increase in accumulated precipitation during the last night with slightly higher values than in N100. In DRY, the convection is slightly more organized than in N100 and thus more efficient in generating precipitation. Since there is a moisture perturbation in DRY, differences between DRY and N100 are also not solely attributable to the indirect effect. Furthermore, AUG03 has the smallest differences (compared to AUG16 and AUG31) in the forcing conditions between the experiments (CTRL, Aer-rad-off, N100 and DRY) which also leads to the smallest differences in precipitation between N100 and the rest of the simulations."

Minor comments:

- I felt that the paper seemed to jump around a bit in terms of the figure discussion. The figures are all referenced in order of first mention starting in Section 3.1, but the actual scientific discussion (Sec 3.2, 3.3) jumps through Figs 5, 10, 2, 5, 11, 7, 2, 11, 6, 5… I understand why the authors have organized the discussion by parameter, and each set of effects is illustrated by different figures so maybe this can't be helped, but it makes the full picture a bit hard to understand without a lot of scrolling. Another suggestion for flow and readability would be to more explicitly label sections 3.3-3.5 with the case differences (i.e., L191-193), to better interpret the relevant differences when jumping around between the different figures (or conversely, somehow identify the panels/curves in the figures by their relevance to e.g. "aerosol indirect effects," which I essentially already suggested in the second comment from the top). But these are just suggestions that might further improve an overall decent paper.

Thanks for the suggestions. We have added to the title of sections 3.3-3.5 the names of the cases (within parenthesis) that we are comparing in each section.

Regarding the order of the figures and the discussion, we appreciate the reviewer's comment. However, we are not convinced that a reorganization of the discussion and figures would improve the readability and flow (a previous version of the manuscript did have the figures organized differently, but this was even less optimal).

- There's some discussion of the definition of SCT being a "soft" reference (L454) but nonetheless a reference that the authors use—I might suggest a vertical line or an arrow to indicate where this threshold is met on Figs 2-4 (but especially Fig 4) in each case.

We think that drawing a vertical line will give the idea that the SCT is something that happens at a specific time in the simulations. In reality, the SCT is a process that happens during a period of time. Therefore we highlight that the SCT metrics found in literature do not fully capture yet the complexity of the phenomenon. For this reason we treat the SCT definition as a "soft" reference and decided not to draw any line to mark specific times for the occurrence of the SCT.

- L175: "due to a combination of…" shouldn't this also include direct effects as "all aerosol effects combined" (L193)?

True, we have fixed this in the text.

- L192: "negligible semi-direct aerosol effect"—why is this, if that's included (excluded) with direct effects in aer-rad-off? Especially since L340 argues that semi-direct effects are more important than indirect effects in (I think) the differencing of those two scenarios.

The aerosol absorption in N100 should be small due to the low aerosol concentrations while in Aer-rad-off the absorption is zero as there is no aerosol absorption. Therefore, the difference in aerosol absorption between Aer-rad-off and N100 should be negligible. This has been clarified now in the text

- L200: suggest to define your SW/LW wavelength ranges here, or at L194.

The following information was added in section 2.1:
"The model is coupled to a version of the four-stream Fu-Liou-Gu radiative transfer model (Fu and Liou, 1993; Fu et al.,1997 and Gu et al., 2003) which uses 6 bands for shortwave (solar spectrum between 0.2 and 5 µm) and 12 for longwave radiation (infrared spectrum between 2200 and 10 cm$^{-1}$)"

- L206: "geostationary"  Fixed
- If it's not too much trouble I might recommend to flip the orientation of Figs 2-4 and 11, so that simulations are rows and dates are columns, to match the layout (dates = columns) used in Figs 5-9.

The orientation of the mentioned figures was flipped. Now, the dates are columns in all the figures.

- L445: "considered that the cloud cover should remain below 50%" I'm not sure I follow this meaning, but I'm not familiar with the particular study being referenced.

We have modified the text in order to facilitate the understanding:
"However, they only considered the SCT to happen when the cloud cover remained below 50%, either during the following 24 hours of simulation or until the simulation end."

- Fig 10: are the MIMICA curves for CTRL, or for the average of all experiments (but Fig 4 suggests changes in clouds among the different scenarios). If it's the latter, I'd suggest adding uncertainties to each date; if it's the former just clarify that in the caption.

Fig 10 shows the MIMICA curves for the CTRL simulations. This was added to the caption of the figure.

- While reading on a mobile device (iOS) I made a comment that I'd like to see ranges on the SEVIRI cloud values in Fig 10, but in opening the .pdf on a PC, I see the authors have already provided this! I'm not sure what quirk of technology made my iOS copy of the figure lose the red shading (even after re-downloading-- maybe something strange with the vector graphics?) but I hope the authors and/or editorial team can ensure this is rendered properly on all devices during final publication. I mention it just because this isn't an issue I've seen before. All that aside, it would be good if the caption would describe what's shown by the shading (1 standard deviation in CLAAS-3?)

We haven't had issues with the visibility of the ranges in Fig 10. We will probably hear from the editorial team if they have a similar problem. We thank the reviewer for the advice.

The shading shows the standard deviation (or spread?) over an area about 500 km wide containing the three trajectories. The information will be added to the figure.

- Throughout: double-parentheses in citation formats (e.g. L26, L60, L212…).  Fixed
- L110: missing period Fixed
- L379: suggest comma after "indirect effect" for list clarity. Done
- L385: Table 4 shows  Fixed

- L404: suggest maybe "conditions" rather than "periods," since it's really just three days. We prefer to keep the word "periods" since there is no definition for how short a period should be. Additionally, the word "conditions" is used earlier, in the same sentence.

- L439: above the MBL caused Fixed

---

## Author Response (AR2)

We want to thank the reviewer for the feedback. Below you will find our response detailing the modifications we have made in relation to the last comments and suggestions.

I have one suggested edit (I don't think this affects the scientific conclusions, but may be important for reproducibility), and noticed a few minor technical corrections:
1) Can the authors clarify the model initialization/trajectory conditions: L150 says initial latitude trajectory is set at 25S, but Fig 1 shows trajectory latitudes moving between 16S and 6S, while Table 1 has latitudes around 11S? Are these all referring to different runs?

Thanks for noticing this. We have added the following sentences to paragraph starting on L160:

"The trajectories pass over Ascension Island (8°S, 14.4°W) between 10:30 and 13:30 UTC on a specific day. In our simulations we only use the section of the trajectories within the 64 hours before and 3 hours after noon (or closest available time) when they pass over Ascension Island (Figure 1 b)."

*Therefore the trajectories we show in Fig 1 don't start at 25°S (the initial section of the trajectories was not used)

Technical corrections:
-- L70: "the moisture... can result in the relatively humid air being entrained"? (extra word or verb tense here?) Fixed
-- L89: instead of -, suggest "and" between the two named campaigns    Fixed
-- L181: in the revised sentence, suggest "the differences are due to a combination of all effects" or "the differences are due to all effects combined" but both words reads as redundant.    Fixed
-- Table 1: specify times are(?) in UTC, in addition to the Latitude comment above (also if Longitude varies as Fig 1 suggests, should that be stated as well?). Times are in UTC. Fixed
-- Fig 3 caption: I think you lost the 1 from AUG31 in the process of revision.    Fixed

Regarding the "Remarks from the preceding review file validation":
We changed the colors in Figures 1, 5, 6, 7, 9 and 10. Now we use colors from the "seaborn-colorblind palette" in python. We also use different line styles to make the lines more distinguishable.